# Inclusive Aging in Korea: Eradicating Senior Poverty

**DOI:** 10.3390/ijerph19042121

**Published:** 2022-02-14

**Authors:** Jooyeon Kang, Jungmin Park, Jaemin Cho

**Affiliations:** 1Goldman School of Public Policy, University of California, Berkeley, CA 94720, USA; pmokang@korea.kr; 2College of Business Administration, University of Ulsan, Ulsan 44610, Korea; jmpark2@ulsan.ac.kr

**Keywords:** senior poverty, aging, social security, re-employment, pension, Korea

## Abstract

Poverty for the elderly is one of the most urgent social problems when discussing the social problems facing Korean society. The purpose of this study is to identify the causes of elderly poverty problems and to seek countermeasures. According to a systematic analysis of the economic difficulties of the elderly population that applies a socioecological model, the cause of elderly poverty is complicated by the specificity of the labor market and pension system in Korean society. This is compounded by the lack of a public support system that can overcome insufficient family care and a lack of individual preparation. To alleviate elderly poverty, this paper recommends three policy alternatives. First, a robust multipillar retirement income security system must be established. To secure a minimal retirement income for the elderly in poverty, who have been marginalized from the public pension system design, the basic pension should be raised for the bottom 70% of senior citizens. Second, in order to tackle labor market duality and early retirement, the seniority-oriented wage system should be reformed into a job-based wage system. Third, to minimize unemployment and promote quality among re-employment jobs, the government should strengthen vocational skills development by expanding programs tailored to older people.

## 1. Introduction

Korea has achieved impressive economic development, referred to as the “Miracle of the Han River”, over the past half-century. Korea overcame absolute poverty by generating strong income growth, but the fruits of this economic growth were not equally distributed. Korea’s old age relative poverty rate is the highest among Organization for Economic Co-operation and Development (OECD) countries. Many experts have suggested that the overwhelming elderly poverty rate proves that income has not been redistributed well between generations and classes [1,2,3,4]. In addition, Korea is expected to enter a super-aged society having more than 20% of aged people among the total population by 2025 [5], which increases the risk of elderly poverty. The theme of this study, the problem of poverty among the elderly, arises at the interface of two key issues in Korean society: polarization and aging. 

The current poverty and income inequality among the elderly are attributed to the fact that inequality in the labor market is reflected in the social security system at a time when private support for the elderly is weak. A socioecological system perspective that focuses on the interaction between an individual and the environments surrounding that individual indicates diverse poverty factors. First, on an individual level, there is not enough preparation for retirement. Although awareness of preparation for retirement has increased, senior citizens fail to prepare for their retirement due to higher spending. This spending often comprises debt repayment and education expenses for children that are more than these citizens can afford. Second, an increase in women’s social participation and nuclear families at the interpersonal level weakens the family’s private support of aged parents. Third, the legal retirement age in Korea is 65, but the average retirement age is much lower due to corporate labor cost avoidance. Fourth, at a community level, Korea’s labor market is divided between a primary labor market composed of large companies and regular workers and a secondary labor market consisting of small- and medium-sized companies and irregular workers. This division results in high income inequality. Workers in the secondary sector often find preparing for retirement difficult by means other than public pensions, which are not sufficient enough. Moreover, retirement pension, one of the representative public pensions, is often not subscribed to, since only workplaces with more than 300 employees have been obligated to introduce retirement pensions since 2019, so most small- and medium-sized workplaces have offered it voluntarily. Finally, in terms of the legal system, Korea’s public pension has a low-level income replacement rate that refers to the pension as a proportion of income received by the worker, and a wide gap that does not include the elderly who worked in the secondary job market or who were unemployed.

Poverty among the elderly is always mentioned as one of the most urgent social problems when discussing the issues facing Korean society. The seriousness of elderly poverty is also evident in the fact that Korea maintains one of the world’s top 10 GDP levels while showing the highest elderly poverty rate among OECD countries. On average, OECD countries do not have much difference in poverty risk by age, but in Korea, poverty risk increases rapidly in old age. The poverty rate of senior citizens in Korea, which saw a median income of 50% in 2016, stood at 43.8%, more than three times that of the OECD national average of 13.5% (Figure 1). 

Along with this prevalent risk of aging poverty, the world’s fast pace of population aging is rapidly increasing the number of elderly people in poverty (Figure 2). If the current poverty rate is maintained, the number of people aged 65 or older is expected to rise by 31.9% in 2025, from 3.5 million to 4.6 million (Table 1). Entering a super-aged society means more that 20% of the population are elderly [7]. Approximately one out of every two impoverished people will be elderly in the future.

Social and economic problems caused by elderly poverty are not just income issues but can affect various aspects of Korean society. Korea has the highest suicide rate among OECD countries [8], and suicide rates increase with age (Figure 3). The 2017 elderly survey conducted by the Ministry of Health and Welfare indicated that their major suicide motivations were financial difficulties (27.7%) and health problems (27.6%). A needy life may lead an individual to commit suicide. If the government does not actively respond to the economic insecurity of the aged generation, social problems caused by the income gap and economic inequality of the elderly are expected to intensify. Therefore, to increase opportunities to age with equity and dignity, this study aims to analyze the current status and causes of elderly poverty in Korean society and to explore future policy responses. 

The study is divided into three domains. First, Section 2 examines the current status and trends of elderly poverty in Korea and analyzes the characteristics of this status and these trends. Section 3 systematically identifies various factors affecting the entry into poverty in the elderly. Section 4 proposes a policy alternative that can protect the elderly from poverty in a long-lived era based on the factors of influence previously analyzed.

## 2. Dynamics of Elderly Poverty in Korea

### 2.1. Definition of Poverty

The dictionary meaning of poverty is a state in which the minimum basic needs required for survival are not met, i.e., a state in which basic material needs cannot be met due to a lack of resources [3]. Here are various definitions of poverty, but there are two main approaches to defining poverty. These are absolute poverty and relative poverty. Absolute poverty is the condition in which household income is insufficient to afford the basic necessities of life. This definition of absolute poverty stems from Rowntree [9] who described “primary poverty” as “households whose total earnings would be insufficient to achieve the lowest level of total income required to maintain physical efficiency”. Discussion of relative poverty began when Townsend [10] criticized this definition because of the lack of consensus on basic needs for survival, explaining poverty in terms of income in relation to the income level of the entire population. Relative poverty refers to a lack of basic ability to achieve minimum living standards. The areas of assessment include proper nutrition, participation in activities, and availability of resources for widely accepted minimum living conditions [11], or for living conditions to reach a minimum acceptable level [12]. 

The poverty line or poverty thresholds are the minimum income levels needed to achieve a proper standard of living in a given country. The poverty line can be divided into the absolute poverty line belonging to the objective poverty line and the relative poverty line according to the measurement method. The absolute poverty line is determined by estimating the cost of purchasing essential goods in a society and considers the condition that does not reach the objectively prescribed minimum standard of living as poverty. The absolute poverty rate measures the proportion of households below the poverty line to all households after establishing a certain standard of poverty. The World Bank’s global poverty line of one dollar ninety cents per day is widely used. However, relative poverty lines are determined in a comparative manner, such as a specific ratio based on average or median income. Townsend [13] defined the relative poverty line as 80% of the average household income and 50% or less of the average household income for the poorest. For international comparison, this study defines poor households as less than 50% of the disposable median income based on modified household income (I_k_) weighted by the OECD [14]. The household equivalence scale is an indicator of the additional rate of income growth required to maintain the same level of welfare as the number of household members increases. The expression of household income, denoted by the equation $\sqrt{n}$, adjusted by the equivalence scale is I_k_/$\sqrt{n}$, where n is the number of household members. In 2020, Korea’s poverty line was KRW 1,499,000 (USD 1249, USD 1 = KRW 1200) per year [15].

### 2.2. Poverty Status by Assets and Income

There is a possibility that the status of poverty is statistically overstated or underestimated due to the measure of relative poverty based on disposable income. For example, people having large assets should be excluded from poor households, even if their income falls short of the poverty line. However, distinguishing the difference with available data is difficult [16]. If the impact of asset retention and income are considered in the calculation of poverty rates at the same time, the relative poverty rate could fall by 17% [1]. However, even if poverty rates do fall, the seriousness of the fall and relative position of elderly poverty in the world are not likely to change. A high proportion of individuals in most countries except for the United States have real estate [17], so their poverty rates would decrease too, when all assets are considered. In addition, property income is reflected in the total disposable income since a household gross income consists of employee income, self-employment income, property income, and transfer income. Concerning property that does not generate income, there are no data that accurately reflect the diversity and complexity of assets [18]. Therefore, the identification and comparison of its impact is difficult. Regarding concerns related to identifying wrong policy targets when implementing policy alternatives, the possibility of confusion in policy targets is low. This is due to the fact that the Korean government operates the allowance and benefits calculation schemes for social security to reflect specific nonfinancial assets including land, buildings, cars, and golf course memberships.

### 2.3. Poverty by Age and Gender

As shown in Figure 4, poverty among the oldest, those aged 75 and over, is more frequent than among the younger old, those aged 65–75 [6]. The poverty rate in the latter group is 35.5%, but the poverty rate in the oldest, aged 75 and older, is substantially higher, 55.9%. Most OECD countries have similar trends where age is proportional to poverty rates: the OECD average poverty rate is 11.6% for the younger old and 16.2% for the oldest. Although Korea also has this tendency, the difference between the two groups is particularly high, showing a 20.4% gap [6]. The main drivers of higher poverty incidence among the oldest are less or no contribution in the National Pension system [1], which was introduced in 1988, and a lower ability to work [19]. The older aged group is less likely to participate in the labor market and expend more existing savings or assets. This increases the risk of poverty and reduces the likelihood of escaping poverty [20].

The poverty rate of Korean male seniors was 37.1%, but that of female seniors was 49%, 11.9% higher (Figure 5). Considering that the OECD average poverty rate for male senior citizens is 10.3% and the poverty rate for female senior citizens is 15.7%, the Korean poverty rate for female senior citizens is particularly serious. Higher poverty risk among women than men is attributed to longer life expectancy and lower earnings-related pensions [6]. This also reflects differences in the past labor market, such as male and female work experiences and the gender wage gap [21,22].

### 2.4. Poverty by Household Type

One of the changes that should be of interest in the issue of elderly poverty is the change in the family structure in old age. Korea was a family-care-oriented society that supported aged parents [23]. This used to offset the weak national care system. However, as the family culture becomes westernized, married children do not live together with their parents. The proportion of the elderly living with their children decreased from 54.7% in 1994 to 23.7% in 2017. Accordingly, the proportion of households composed of only the elderly, such as one elderly person or elderly couple, is increasing. The proportion of only the elderly households surged from 40.4% in 1994 to 55.0% in 2004, reaching 72% as of 2017 [24,25]. Looking at the household types of all the elderly aged 65 or older specifically, the distribution was 23.6% living alone, 48.4% living with an elderly partner, 23.7% living with their children, and 4.4% other [26]. According to the relative poverty rate by types of households, single elderly households had the highest poverty rate, 71.5%. This was followed by elderly couples, other elderly households, and non-aged households: 51.4%, 28.7%, 14.3%, and 10.2%, respectively (Figure 6). The proportion of single elderly households is expected to continue to increase in the future due to the differentiation of households between parents and children. Establishment of a public support system for senior citizens who have difficulty feeding themselves or do not have anyone to care for them is necessary to ensure a minimum standard of living in response to the decrease in private care by their children.

## 3. Drivers of Senior Poverty in Korea 

### 3.1. Framework: Socioeconomic Model

There are various competing theories about the causes of poverty. The debates include advocates who support cultural/behavioral arguments or structural/economic arguments [27]. However, poverty among the elderly in Korea cannot be explained by just one point of view, and various factors have affected the poverty level over the long term. This study seeks to analyze the causes of poverty by applying socioecological models, a theoretical framework useful for understanding multidisciplinary levels of factors affecting individual behavior. Socioecological models built from Bronfenbrenner [28] have been used primarily in the field of public health. The key principle of this model is that various dimensions, including individuals, between individuals, organizations, communities, laws, and policies mutually influence behavioral status [28,29,30,31]. Considering the fact that there are various factors to explain poverty, approaching poverty systematically is feasible. This includes the consideration of all diverse levels of external environments in which individuals belong, rather than only focusing on a personal level of responsibility [30], such as an insufficient preparation for retirement. In addition, socioecological models can effectively shed light on developing policy improvement measures, including not only policy targets but also social practices and cultural and legal environments [31]. The socio-ecological model application on drivers of senior poverty is shown in Figure 7.

### 3.2. Key Factors: What Explains Poverty among the Elderly

#### 3.2.1. Individual Level: Retirement Preparation

Each person’s economic preparation for retirement affects their retirement poverty. Generally, retirement is a predictable event, but economic retirement preparation behavior is different from person to person. According to the Social Security Public Perception Survey conducted by the Ministry of Health and Welfare in 2020 [32], 57% of the respondents said they were preparing for retirement; the remaining 43% said that they were not preparing for retirement. Preparing for retirement differed greatly depending on the type of housing occupancy. The more unstable the housing occupancy contract type was, the lower the percentage of respondents who said yes about the question of whether they were preparing for retirement. The response that they were preparing for retirement was 64.7% for self-ownership, 47.2% for lease on a lump sum deposit basis, and 21.8% for monthly rent. The higher or more stable the salary level, the more people were prepared for retirement. For example, 38.9% of temporary and daily wage workers and 30.4% of unemployed workers said that they were preparing for retirement, while 56.9% of regular wage workers and 66.8% of employers/self-employed and unpaid family workers said that they were preparing for retirement. The reason the cause of poverty in old age cannot be attributed simply to individual lack of preparation is that, even if self-responsibility is high, implementation of economic retirement preparation is difficult if the employment status is low, or the housing type is unstable due to a lack of economic resources. Specifically, current seniors in their 70s and 80s were born in the Japanese colonial era. These individuals lost their families, fled during the Korean War, and worked hard to overcome absolute poverty after the war. Then, most of them lost their jobs during the 1997 financial crisis. Considering Korean development history and their contribution, blaming them for the present poverty of the elderly is unfair.

#### 3.2.2. Interpersonal Level: Changes in Filial Obligation Norms of Family Caregiving

Changes in family structure can also be linked to poverty [33]. The weakening of family support is an interpersonal factor affecting poverty in the elderly. As mentioned earlier briefly, changes in the structure of the private support system can be seen through changes in the form of households. As the number of elderly parents and adult children living together has decreased, the number of elderly couples and elderly households living alone has increased. This means that the elderly population is being separated due to the aging population of the nuclear family as the result of industrialization. In particular, the child support responsibility for bereaved parents in Korea, especially that of the daughter-in-law, was strong. However, the increasing number of working couples due to women’s social participation also contributed to the collapse of the private support system. 

In addition, the attitude of the elderly toward supporting their old age is rapidly changing. This change in the value of parental support of children can be explained as a change in values and attitudes toward parental support. Familism centered on filial piety has weakened and social norms and systems have changed. According to the Statistics Korea survey (Figure 8), 70.7% of the respondents answered that they were family members in 2002, but only 22% responded that way in 2020; and 61.6% said the government and society should be responsible for parental support together.

#### 3.2.3. Organizational Level: Early Retirement and Seniority-Based Wage System

Early retirement from jobs reduces lifetime income, adding to the risk of poverty in old age. According to the Senior Employment Promotion Act, early retirement of middle-aged and older workers in their late 40s and early 50s has been considered honorary retirement. According to the 2019 Economically Active People Survey conducted by Statistics Korea [36], the age at which the elderly (55 to 64 years old) in Korea quit their longest-serving jobs continues to fall from 50.3 in 2006 to 49.4 in 2019. 

The main driver for early retirement is the salary pay system. The seniority wage system is a structure in which the basic salary continues to rise depending on the number of years of service. The rigidity of the corporate wage system centered on the salary system is the main reason for the avoidance of employment of middle-aged and older people, early retirement pressure, and the spread of irregular workers [37]. Recently, performance-based wage systems such as the annual salary system have been spreading, but the salary-based wage system is still dominant [38]. In the salary-based wage system, wages rise in proportion to the number of years of service, resulting in a wage gap according to the number of years of service. As employees’ work productivity is not simply a result of their long service years, the aging and long-term service of employees who receive higher wages but fall short of productivity lead to an increase in cumulative labor costs that companies have to pay (Figure 9). Companies have a strong incentive to address such problems through a personnel management strategy that includes early retirement, which can slow this increase in costs. In the face of an aging society, there is a need to reorganize the wage system in a way that helps stabilize employment for the elderly.

Another problem with early retirement from major jobs lies in the growing gap between the age of retirement and the age of receiving benefits from the National Pension Service, a public retirement income security system. The pension age varies based on the year of birth. The pension can be received at 60 or younger for those born in 1952 and at 65 for those born in or after 1969. The “income vacuum” can exist for five to 15 years between retirement from work and the age of receiving the pension. In addition, senior citizens continue to work even after officially retiring from the labor market because public pensions alone are insufficient for maintaining their lifestyles. Korea displayed the highest average effective age of labor market exit, 72.3 years, for both women and men (Figure 10). However, labor opportunities are only available for a small number of middle-aged and older people, and the provision of stable labor opportunities with a decent wage is more limited. As experience and proficiency are important factors in professional expertise [39], the shorter the working period in life-long jobs, the more likely the engagement in low-wage work. 

#### 3.2.4. Community Level: Labor Market Duality

At the community level, the gap in workers’ income in the period of economic activity widens the gap in a life after retirement. The total income of households consists of earned income, self-employment income, asset income, and private/public transfer income. Employment income accounts for 64% of the total income (Figure 11).

Earned income varies substantially depending on the size of the enterprise and type of employment (Appendix A). The labor market in Korea is divided according to the size of the workplace and the type of employment [4]. Full-time employees of large companies working in the primary market enjoy high wages, job security, organized labor unions, benefits, and the protection of labor market regulations. However, workers in the secondary labor market are mostly nonregular workers in small- and medium-sized enterprises. These workers suffer from low wages and job insecurity, and their jobs are often not protected by labor regulations [41]. The monthly wage level of nonregular workers is 45 percent of that of regular workers (Table 2). The hourly wage of nonregular workers was improved to 70% of that of regular workers through the minimum wage hike, but the total income gap was not significantly improved [42]. The reason for the gap between regular and nonregular workers is the seniority-based wage system [4]. Regular workers are guaranteed to continue to receive wage raises proportional to their increased work experience, while irregular workers are not.

Based on data from 300 employees, the wages of small- and medium-sized company workers were 51% of those of large company employees in 2016. This barely increased to 53% in 2019 (Table 3). The dominant conclusion is that the wage gap by firm size stems from the productivity gap between large- and small-to-medium-sized companies. This productivity gap is often the result of unfair subcontracting practices [43].

The wage gap between men and women has improved, but women’s hourly wages are still under 65% of men’s (Table 4). This is mainly because women’s careers are shorter than men’s due to career breaks such as those associated with marriage and childbirth. Therefore, women are at a disadvantage when considering wage increase opportunities [21].

Corporate size and employment type duality create income inequality for workers and make retirement planning difficult. For example, secondary sector workers’ limited savings, insurance and pension subscriptions are frequently insufficient to develop a viable retirement plan.

#### 3.2.5. Systemic Level: Gap in the Multipillar Income Security System

Korea’s multilayered income security system, consisting of basic pension, national pension, and retirement pension, appears to be adequate (Figure 12), but a wide gap exists in public pension system entitlements. A poverty-free life after retirement is not fully guaranteed. According to a survey conducted by the National Pension Service [44], the retirement income required for single households was KRW 1.3 million per month and KRW 2.1 million per month for couples. However, only 8.41 percent of single households (15.95% male, 0.79% female) facing retirement received KRW 1.3 million in income. These single households earned a public pension that averaged KRW 520,000. 

The National Pension Service was introduced in 1988 for businesses with up to 10 employees and was expanded to cover all citizens in 1999. The National Pension Service serves as a major retirement income source in the absence of postretirement earned income. Korean subscribers between 18 and 60 years of age pay part of their income as insurance premiums to the National Pension Service. However, since the National Pension Service is applying a reserved method that receives as much as individual contributions, it cannot receive benefits from the retirement income guarantee if the contribution is low or if it fails to join the national pension. Only 41.5% of the population aged 65 and older subscribe to the national pension [19]. Currently, 4.67 million of the 21.84 million subscribers have suspended or long-term arrears due to economic circumstances, and 8.95 million have not joined the national pension because of the lack of participation in economic activities. In total, 42% of the population aged 18–59 does not pay any national pension premium at some point in time. The National Pension Service is the system with the lowest threshold for subscription and the greatest benefit against insurance premiums throughout all pensions, and it is practically difficult to use other private systems if they are shunned even from such a public social security system.

Figure 13 shows the income replacement rate of mandatory pensions in OECD countries. These figures reflect future reforms set by each country, not presently, but by 2060. As the national pension system matures, the pension amounts that new subscribers receive in the future will rise, but the existing gaps will still exist. In addition, the national pension was designed based on an income replacement rate of 40% based on an average lifetime income. However, this is the case when the national pension is maintained for 40 years and premiums are paid normally. If calculated as the replacement rate of real income guaranteed based on payments of more than 10 years, the minimum subscription standard was only 22.4% in 2020 [45]. The ILO and OECD recognize an income replacement rate of 50% as an adequate level, while a replacement rate of 70% is recommended according to the OECD [46].

Basic pensions were introduced in 2008 to support stable retirement income security for older people who do not join the national pension scheme or do not receive sufficient pensions (Table 5). Since 2014, the basic pension amount and application target have gradually expanded. Since 2021, the basic pension provides up to KRW 300,000 per month depending on income to senior citizens aged 65 or older. Unlike the National Pension Service, which requires insurance payments, if their income recognition amount is less than the recipient’s selection standard, they could be a basic pension recipient without contributing to the payment.

The retirement pension system is a system in which employers accumulate funds during the worker’s tenure for retirement benefits. This guarantees that the worker will receive a retirement income that allows for economic stability. Retirement pensions reflect gaps in the dual structure of the labor market. For example, 88.1% of employees of businesses with 300 or more full-time workers have retirement pensions, and 23.5% of employees of small- and medium-sized companies with less than 30 employees have retirement pensions [48]. Regular workers had a 57.2% retirement pension subscription rate, while only 23.2% of nonregular workers were retirement pension subscribers [42]. The retirement pension system, which is currently required only in workplaces with more than 10 employees, will be mandatory even for workplaces with less than 10 employees in 2022. However, since the lump sum receipt rate is very high, 97.9% as of 2018, and the number of early withdrawals has increased (36.6% increase in 2018 compared to 2015), there is a limit to the universal retirement income guarantee function [49].

## 4. Policy Recommendations

To tackle the aforementioned social problems, the South Korean government has pushed to expand the number of people eligible for a basic pension, support national pension premiums, ease the qualifications of the basic living security system, raise the minimum wage, and make nonregular workers regular. However, more effort is still needed to solve the complex problem that has accumulated over a long period of time.

### 4.1. Reinforcing Zero-Floor of the Multipillar Pension System

As of 2021, Korea’s elderly population ratio is 16.6%, but the level of public spending on welfare for the elderly is 2.8% (2019), which is very low compared to the OECD average of 7.4% (Figure 14). Major OECD countries used 7% of the GDP for senior citizens’ income security spending. Senior citizens in these countries averaged around 14% of the total population [50]. Therefore, Korea needs to expand the government’s financial role to address economic difficulties in old age. 

First, it is recommended to strengthen the pay adequacy of a multilayered income security system, focusing on basic pensions. Korea operates a national pension, basic pension, and retirement pension to support postretirement aged workers. As the first and foremost solution, the basic pension, pillar 0 or the pension floor should be raised to guarantee that the poor elderly can have lives that meet minimum quality of life standards. This applies to those who are unable to earn an employment income or to expect a private/public transfer [52]. Most of the current generation of senior citizens was unable to join the national pension system when the system was introduced. This system is key to guarantee retirement income. Some senior citizens may have been able to join the system, but their contributions were too low to secure benefits. However, the basic pension is paid to the bottom 70% of the elderly. This pension plan has the potential to have a strong antipoverty effect by helping the lower economic class of elderly with fewer national and retirement pensions. The basic pension started at KRW 80,000 in 2008 and now stands at KRW 300,000. This allowance currently accounts for only 12% of the total income of national pension subscribers. In 2020, the average income of these subscribers was KRW 2,539,734 [53].

Considering that the labor market history reflects the national pension, retirement pension, and elderly poverty, basic pension redistribution should have a stronger antipoverty role. However, although basic pensions have practical poverty alleviation effects, the cost may be unacceptable to central and local governments. Raising the basic pension by KRW 100,000 will cost KRW 932 billion in 10 years and will cost an additional KRW 4997 billion to cover the elderly aged 75 or older (Table 6). These expenditures are open to the criticism that such huge spending will affect other social welfare programs by pressuring central and local government’s finances. 

A public awareness survey conducted by the Ministry of Health and Welfare last summer found that the public agreed to make social security spending for the elderly a top priority [54]. Regarding the priority support age group for welfare policies, 35.9% of respondents answered that the elderly was the priority for welfare policy, followed by middle-aged and elderly (24.5%) and young (19.5%). Experts also said support for senior citizens (47.0%) was the first priority. The public prioritized the elderly (33.3%) with single parents and grandchildren (25.6%) and those with low incomes (23.4%) to receive support at a similar level. These findings reflect the possibility of intergenerational solidarity to solve poverty for the elderly. 

There may be those with the counter argument that individuals will avoid paying the national pension in order to be eligible for the basic pension. However, the prospect of evading the national pension seems excessive because of fears associated with elimination from the basic pension list. In addition, the current national pension is a mandatory system. Premiums are automatically deducted from an individual’s monthly salary. This ensures that those with an income cannot avoid paying the national pension. 

#### National Pension and Retirement Pension

A plan could also be implemented to raise the income replacement rate of the National Pension Service, which is currently 40 percent, to increase public pension benefits for senior citizens. However, while adjusting the income replacement rate of the national pension may be effective in preventing future generations from poverty [55], this adjustment does not solve the poverty problem of the current generation of senior citizens who are already in the national pension gap. Specifically, the adjustment cannot solve the poverty of the current generation of female senior citizens who had no labor participation in the past. As for the national pension gap, the government is currently providing national pension premiums to low-wage workers working in small businesses with less than 10 employees and farming and fishing villages (Durunuri Social Insurance Support Project) [56]. In addition, the government operates a credit system that recognizes the payment period of insurance premiums to those who cannot continue to pay the national pension due to childbirth, military service, and unemployment.

For the middle class, the government should further strengthen the use of retirement pensions to cover old age income. To this end, retirees should be induced to receive these in the form of pensions, not lump sum payments. The retirement pension is a legal obligation system applied to workers employed for at least one year, and the user shall bear the full financial resources. In the reality of Korea’s labor market, not many people are eligible for retirement pensions; retirement pensions are not universal pensions that cover many classes of workers. Nevertheless, most middle- and upper-class workers are bound to receive retirement pensions and can, therefore, play a role in the multilayered pension system by class. The key task at this time is to establish retirement pensions as pensions, not lump sum payments. To this end, the collection of lump sum payments may be restricted by designating mandatory receipt of retirement pensions for a certain period of time or by strengthening tax support for the receipt of pensions. The reason that retirement pensions are often received as lump sum payments is the resemblance to the severance pay tradition. There is a considerable demand for lump sum payments. If an individual retires in their 40s and 50s, the individual is not old enough to leave the job market yet. Lump sum payments may be necessary to provide an initial investment for self-employment. According to the 2019 Statistics Korea survey [49], 58.6% of the respondents were in favor of mandatory retirement payments in the form of pensions, and 36.4% were against the idea. In the future, when the retirement pension system matures, individual reserves will have increased, increasing the effectiveness of the pension. This, in turn, could further increase the role of retirement pensions for life after retirement. 

However, some maintain that conditions for joining should be eased so that workers under one year can receive retirement pensions in order to resolve the gap in retirement pensions [57]. In 2016, a partial amendment to the Workers’ Retirement Benefit Guarantee Act, which calls for paying retirement pensions to workers for more than a month or less than a year, was proposed in the National Assembly but abolished in 2020 due to the expiration of the National Assembly session without sufficient social discussions [58]. Given current employment conditions, relaxing the conditions for membership may result in reducing existing jobs rather than achieving the purpose of protecting workers despite its potential effect to reduce blind spots in the employment safety net. Large companies will be less affected, but small businesses and the self-employed could responds to the change by reducing employment rather than paying retirement benefits for one-month workers. According to the 2020 Employment Insurance Statistical Yearbook [59], more than 50% of retirees with less than one year of work worked for companies with less than 30 employees. In the face of the economic crisis caused by COVID-19, the burden of severance pay could result in a reduction of low-wage jobs, deepening poverty in the elderly population.

### 4.2. Implementing the Principle of Equal Pay for Equal-Valued Work

#### 4.2.1. Direction to Job-Based System

In order to solve low-income problems caused by the early retirement of elderly workers and the dual structure of the labor market, the same wage principle of equal value labor should be realized. Currently, those who do the same job have a large wage gap. The wage depends on whether the employee is a regular or nonregular worker and belongs to a large company or to a small- and medium-sized subcontractor. Therefore, realizing equal wages for equal value labor is an important task for Korean society. Once employees enter low-wage jobs, they become highly dependent on them. Elderly workers who have received low wages even in life-long jobs are likely to suffer from poverty with fewer wages after retirement [60]. According to a survey conducted by the Seoul Metropolitan Government in 2020, 22.8% (4.9 million) of elderly people living in Seoul and re-employed in full-time jobs after retiring from their main jobs do not receive the legal minimum wage. 

To apply the “Equal Pay for Work of Equal Value” principle, the wage system should be shifted to be job-oriented rather than seniority-driven. A job-centered wage system is a wage system that gives a job rating based on the relative job value and determines the relative level of wages [60]. This is effective in preventing unreasonable wage discrimination because wages depend on the importance and difficulty of the job [61]. 

In order for job pay to succeed, solidarity and bargaining power of all workers must be supported [62]. In Europe, where job pay is common, unions and user organizations set wages for each job or proficiency through collective negotiations by industry. As this decision was applied to all workers in the industry, equal wages for equal labor could be carried out beyond the corporate fence. For example, Germany’s job pay system is negotiated directly by the head of the labor union and the head of the government. These negotiations determine the wage structure and wage level according to the industrial group agreement [63]. Because post adjustment of individual workplaces is not possible, the wage gap by institution or employment type is not large. 

However, Korea’s trade union organization rate is only 12.5 percent (2019), and more than 90 percent of workers have no organization to represent their interests [64]. In addition, the gap between the union organization rate and the number of subscribers depending on the size of the workplace is large (Figure 15). The wage rates negotiated by organized labor unions are significantly higher than those that were determined by the employer alone. 

#### 4.2.2. Advocates for All Employees: The Chamber of Labor

Therefore, consideration for establishing the labor chamber as a way to improve the bargaining power and representation of the entire workforce is necessary. All workers with a certain period of employment insurance payments, including irregular workers, special types of workers, and the unemployed, are obligated to sign up to the chamber. The Chamber of Labor, a labor representative organization corresponding to the Korea Chamber of Commerce and Industry, can be in charge of job-oriented wage negotiations at the national level, not at the company level [66]. In addition, the Korea Chamber of Commerce and Industry and the Labor Chamber may set up the “Korea Labor-Management Committee,” a central government-level labor-management relations organization to discuss labor and industry-related policies and to lay the foundation for social dialogue and compromise. 

The Austrian Chamber of Labor could serve as a reference model for the Korean Chamber of Labor. The Austrian Chamber of Labor was established in 1920 as a counterpart to the Chamber of Commerce [67]. The most important functions are to participate in the process of revising labor- and social-security-related laws, presenting positions on law proposals and improvement measures, and to represent labor interests in policy directives. The labor chamber is a “think tank” involving experts from various fields, including lawyers and labor, as well as workers. The labor chamber conducts about 400 collective agreement negotiations every year. In addition, the company carries out legal advice, education, and consumer protection. Unlike labor unions in which membership is autonomous, private-sector workers are obligated to join ensuring excellent worker representation. The Chamber of Labor established in nine metropolitan governments gather to form the Federal Chamber of Labor; and the Vienna Chamber of Labor, in the capital city, is in charge of the chairman’s organization. The Chamber of Labor represents a total of 3.6 million workers, excluding civil servants and agricultural workers, and the interests of retired workers are represented through the Chamber of Labor. Finance is covered by the Labor Chamber Levy, an amount equal to 0.5 percent of the pretax wage. 

There may be resistance from some unions to the introduction of the labor chamber, but effective measures are urgently needed considering the fact that there is no organization representing the ever-expanding interests of vulnerable workers. In response to this, a central labor-management organization called the Chamber of Labor could be effective in easing the dual structure of the labor market. This requires that regular unions of small and large companies overcome the problem by representing the entire labor community and form solidarity with super-business agreements to switch to job-based pay structure.

### 4.3. Developing Knowledge and Skills for Transitions

Alternatives are needed to minimize unemployment and allow re-employment with decent jobs in response to early retirement. One option is to increase productivity by providing vocational training to retirees o1r prospective retirees. Generally, negative stereotypes and prejudices that older people are less productive are common [68]. Most businesses are not active in vocational training for older workers because the cost-to-return ratio is perceived to be lower and less effective than for younger workers [69]. In particular, the need to acquire work-related knowledge or skills is not high because the older workers are usually in charge of labor-intensive low-skilled tasks when working in the secondary sector. As a result, workers’ career development opportunities are limited and affect early retirement and re-employment. The probability of transition from nonregular to full-time employment is less than 2%. In addition, most of the re-employment jobs are nonregular to economically inactive or shift to low-skilled simple labor jobs such as security guards and street cleaners [70].

#### Promoting Vocational Learning System by the Elderly-Customized Program

There are employment education centers for senior citizens specialized in vocational training, but there are only 15 centers nationwide. This is insufficient to meet the needs of workers aged 50 and older, and most of the centers’ programs are focused on simple work such as cleaning and janitorial work [71]. To improve middle-aged and elderly employment capabilities, more professional and effective vocational programs should be developed such as education and training linked to primary life-long jobs, education and training needed for those who want to change careers, education that informs the worker on what services the government provides for those middle-aged and older, and advanced computer literacy education [72]. The demand for employment education also varies because of the large differences in physical function, educational background, experience, and available time. In order to properly respond to these situations, the expansion of various forms of customized vocational training education programs is necessary. The most significant factor that causes the current low participation rate in vocational training for senior citizens is that vocational training is not linked to employment [73]. Developing demand-tailored education and training programs by regularly examining companies’ workforce needs can increase the possibility of employment in a decent job and reduce poverty in the elderly.

## 5. Conclusions

Despite the government’s steady efforts, the relative poverty of the elderly population remains unresolved. According to a systematic analysis of the economic difficulties of the elderly population that applies a socioecological model, the cause of elderly poverty is complicated by the specificity of the labor market and the pension system in Korean society. This is compounded by a lack of a public support system that can overcome insufficient family care and a lack of individual preparation. First, the national pension, which is the core of public pensions, has a wide gap and insufficient pension receipts due to the immaturity of the system. Most retirement pensions are received as lump sum payments and lose their function as pensions. Second, the duality of the labor market creates income inequality by size, employment type, and gender and has a negative impact on the subscription and maintenance of multilayered pension systems. Third, the seniority wage system leads to early retirement from primary jobs in life, which generates an income crevasse until the time of receiving the public pension. 

To alleviate elderly poverty, this paper recommends three policy alternatives. First, a robust multipillar retirement income security system must be established. To secure minimal retirement income for the elderly in poverty, who have been marginalized from the public pension system design, the basic pension should be raised for the bottom 70% of senior citizens. Even if the national pension system, the core of the multilayered income security system, matures, the existing wide gaps cannot be resolved, so strengthening the social safety net will be a feasible alternative to elderly poverty. The role of the retirement pension should also be strengthened as an income source by granting tax benefits to the pension recipient to reduce lump-sum payouts. Second, in order to ease the dual structure of the labor market and prevent early retirement from jobs, the seniority-oriented wage system should be reorganized into a job-based wage system. In order for wage structure transition reform to succeed, a Chamber of Labor, an institution that can represent the interests of all workers, is necessary. The ultra-corporate collective bargaining process centered on the Chamber of Labor can contribute to reducing unfair wage dispersion. Third, to minimize unemployment and maintain re-employment job quality, the government should advance vocational skills development accounts by expanding programs tailored to older people. This study contributes to identifying the causes of elderly problems facing our society and suggests ways to solve those problems. The study, though, used the national level data to provide a snapshot of senior poverty in Korea. Future research could consider comparing regional characteristics, determinants, and trends between rural and urban areas. Given the fact that in rural areas of Korea, the proportion of the elderly population is higher than that of urban areas, the problem of elderly poverty may appear more dramatic in rural areas. This approach could provide more sophisticated targeted policy measures to reduce poverty among the elderly in Korea. For the elderly to escape poverty on their own is practically challenging, if not virtually impossible. In order to realize an inclusive society, the government’s social security policy and labor structure need to be reformed in concert with individual efforts. 

## Figures and Tables

**Figure 1 ijerph-19-02121-f001:**
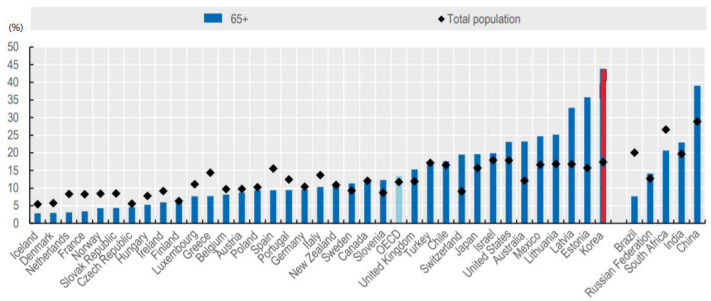
Poverty rates among older age groups and the total population (2016). Source: OECD [6].

**Figure 2 ijerph-19-02121-f002:**
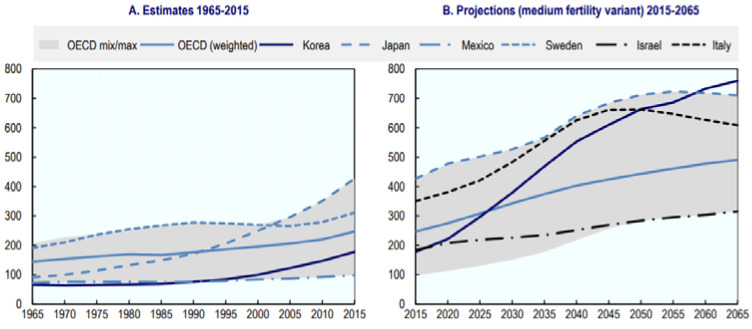
Old age dependency ratio (1965–2015, 2015–2065). Note: number of people aged 65+ for every 1000 of the working-age population (ages 15–64). Source: OECD [6].

**Figure 3 ijerph-19-02121-f003:**
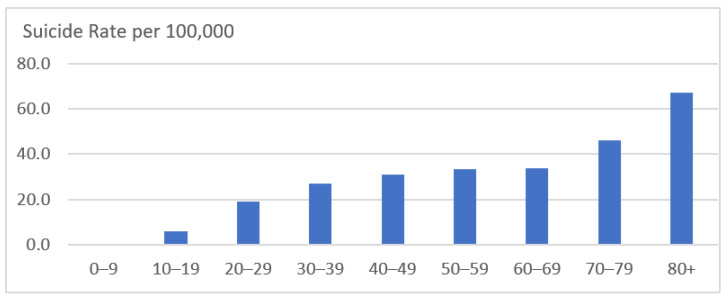
Suicide rate by age group (2019). Source: Statistics Korea [7].

**Figure 4 ijerph-19-02121-f004:**
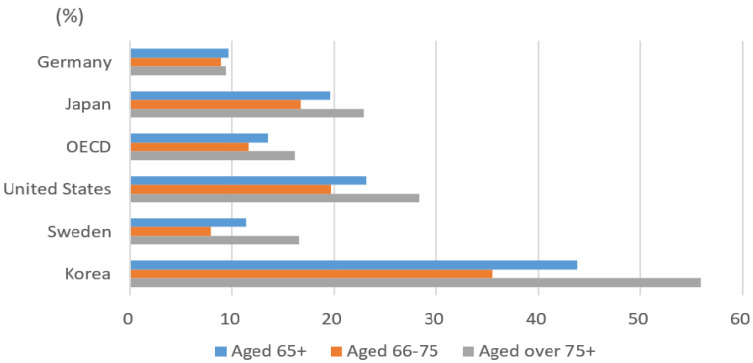
Poverty rate among older age groups (2016). Source: OECD [6].

**Figure 5 ijerph-19-02121-f005:**
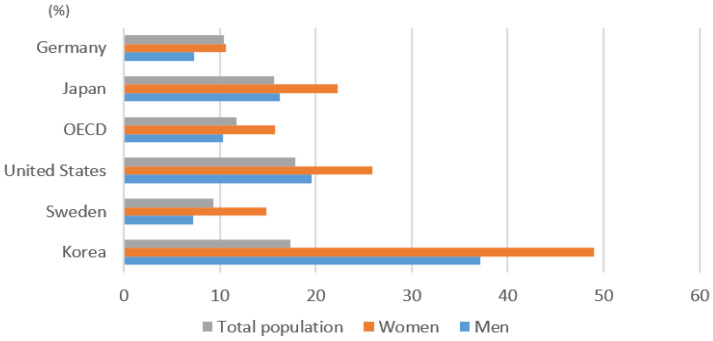
Poverty rate by gender (2016). Source: OECD [6].

**Figure 6 ijerph-19-02121-f006:**
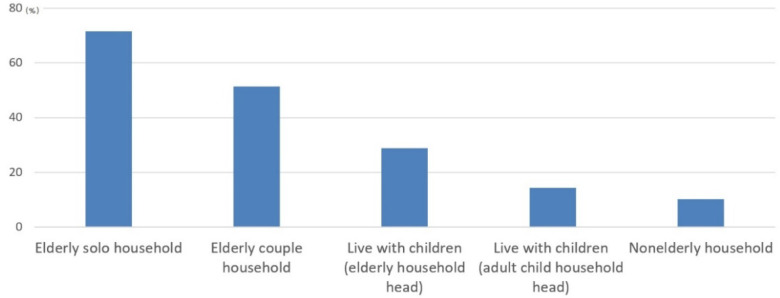
Poverty rate by household type (2016). Source: Korea Statics [26].

**Figure 7 ijerph-19-02121-f007:**
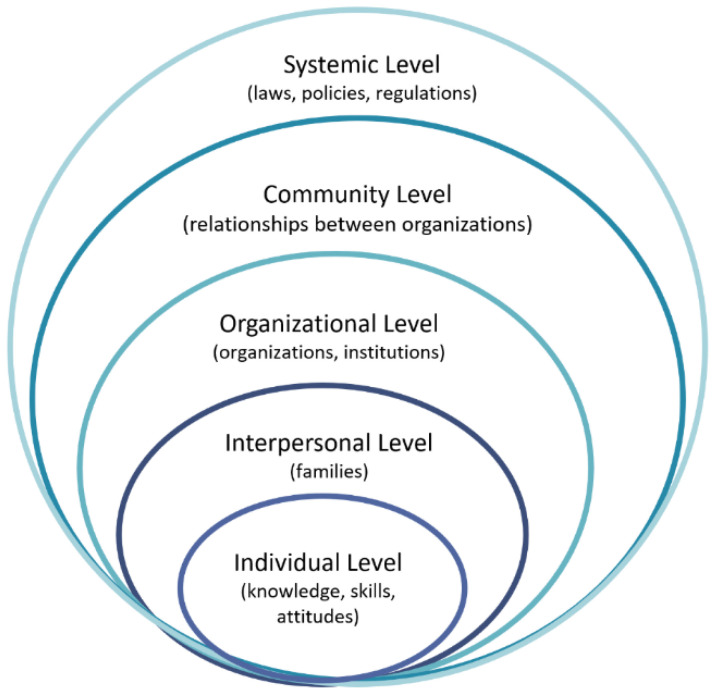
Socioecological model. Source: Concept from McLeroy et al. [29].

**Figure 8 ijerph-19-02121-f008:**
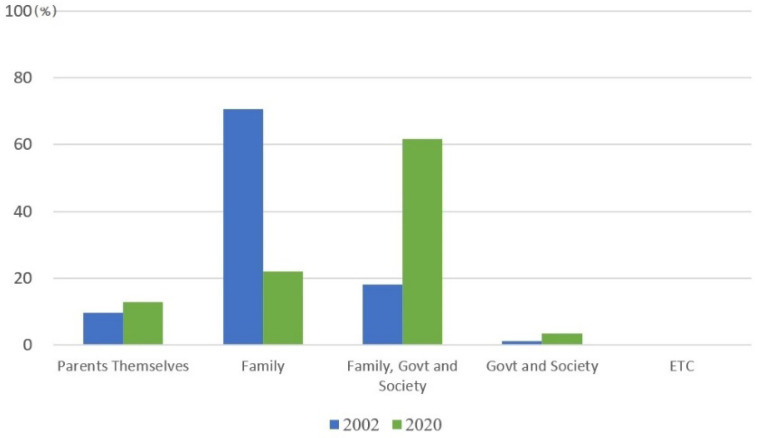
Views on parental support responsibility (2002, 2020). Note: respondents were aged over 15. Source: Korea Statistics [34,35].

**Figure 9 ijerph-19-02121-f009:**
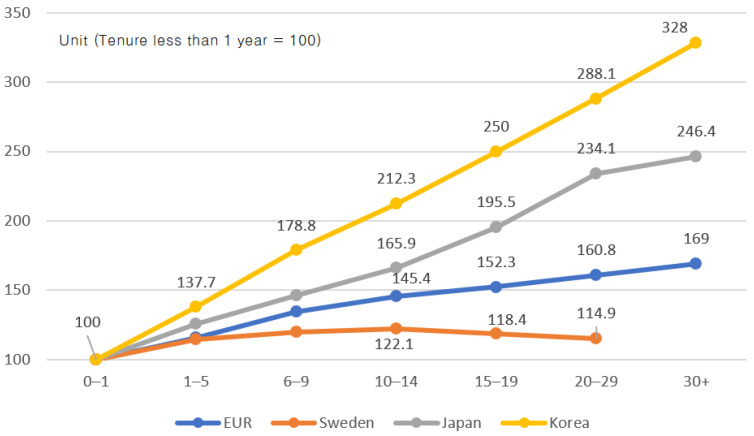
Relative wage dispersion by tenure (2010). Source: Jeong. J.H. [38].

**Figure 10 ijerph-19-02121-f010:**
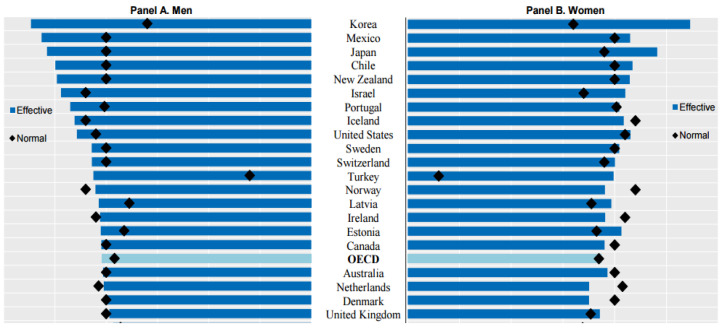
Average effective age of labor market exit and normal retirement age (2018). Source: OECD [6].

**Figure 11 ijerph-19-02121-f011:**
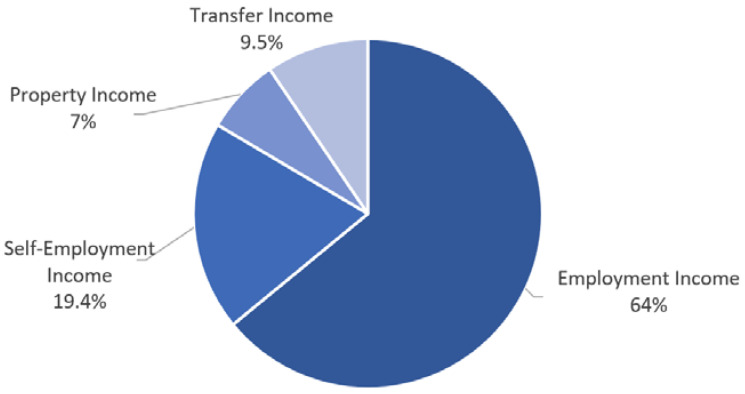
Share of gross household income by source (2019). Source: Statistics Korea [40].

**Figure 12 ijerph-19-02121-f012:**
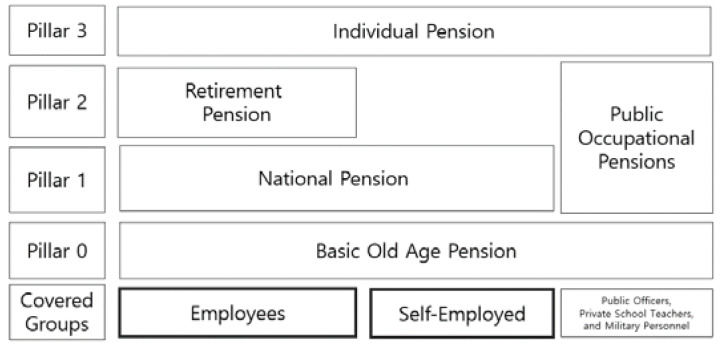
Multipillar old age income security system. Source: Ministry of Health and Welfare [47].

**Figure 13 ijerph-19-02121-f013:**
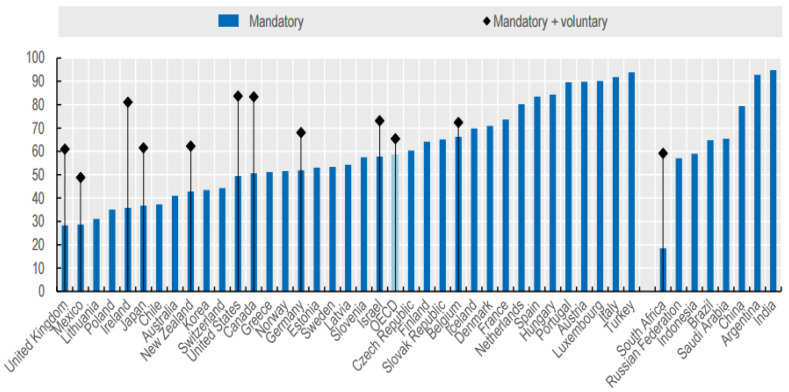
Future net replacement rates for full-career average-wage workers. Source: OECD [6].

**Figure 14 ijerph-19-02121-f014:**
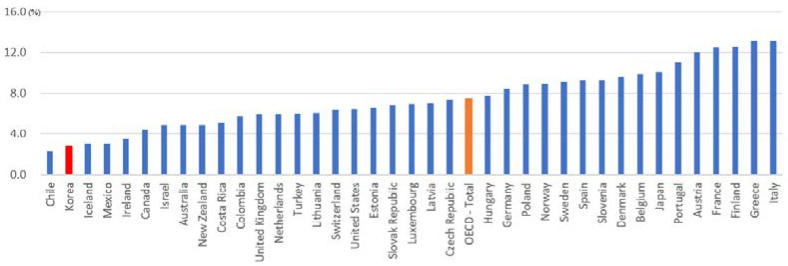
Public social expenditure as a share of GDP (2019). Source: OECD [51].

**Figure 15 ijerph-19-02121-f015:**
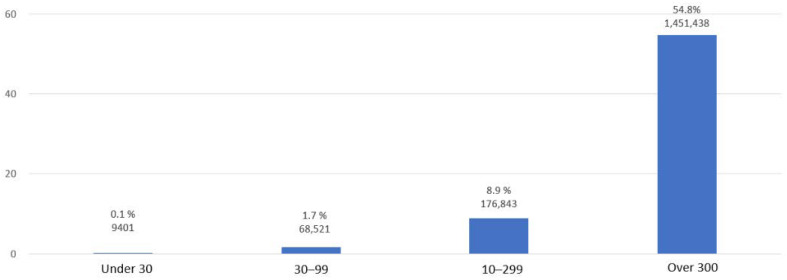
Trade union organization rate and number of members by firm size (2019). Source: Ministry of Employment and Labor [65].

**Table 1 ijerph-19-02121-t001:** Estimates of the total population and the elderly in poverty (2019, 2025).

Object	2019	2025
Total Population	51,849,861	51,447,504
Population Aged 65+	8,026,915	10,585,254
Total Population in Poverty	8,451,527	8,385,943
Population in Aged 65+ in Poverty	3,467,627	4,572,829
Total Poverty rate	16.3%	Status quo
Elderly Poverty rate	43.2%	

Source: calculations from Korea Statistics Data [5].

**Table 2 ijerph-19-02121-t002:** Monthly gross income by employment unit: KRW 1000, %.

Employment Type	2016	2017	2018	2019
Regular Worker (A)	3283	3363	3510	3612
Non-Regular Worker (B)	1445	1506	1588	1643
Share (B/A)	44.0	44.8	45.2	45.5

Source: Ministry of Employment and Labor [42].

**Table 3 ijerph-19-02121-t003:** Monthly gross income by firm size. Unit: KRW 1000, %.

Employment Type	2016	2017	2018	2019
Under 5	1696	1745	1848	1936
5–29	2546	2654	2798	2893
30–299	3125	3244	3400	3478
Under 300 (A)	2510	2577	2706	2794
Over 300 (B)	4954	4902	4964	5258
Share (A/B)	50.7	52.6	54.5	53.1

Source: Ministry of Employment and Labor [42].

**Table 4 ijerph-19-02121-t004:** Monthly gross income by gender. Unit: KRW 1000, %.

Gender	2016	2017	2018	2019
Male (A)	3364	3433	3569	3682
Female (B)	2040	2112	2259	2371
Share (B/A)	60.6	61.5	63.3	64.4

Source: Ministry of Employment and Labor [42].

**Table 5 ijerph-19-02121-t005:** Public pensioners among individuals aged 65 and over.

Year	Population Aged 65+ (A)	Recipients of Old Age Pension and National Pension (B)	Old Age Basic Pensioners without National Pension (C)	Proportion	
				B/A	C/A
2011	5,700,972	915,543	2,902,643	0.16	0.51
2015	6,771,214	1,444,286	3,050,897	0.21	0.45
2018	7,638,574	1,957,696	3,168,035	0.26	0.41

Source: National Pension Service [19].

**Table 6 ijerph-19-02121-t006:** Estimated additional cost of raising the basic pension amount. Unit. 10 billion KRW.

Year	Status Quo (KRW 300,000)	Additional Cost of Raising the Basic Pension Amount to KRW 400,000	
		Lower 70% Income Bracket Aged 65+	Lower 70% Income Bracket Aged 75+
2020	16,921	-	-
2025	23,017	7672	3109
2030	27,960	9320	3819
2035	32,229	10,743	4997
2040	35,689	11,896	6124
2045	37,376	12,459	7020

Source: Korea Institute for Health and Social Affairs [52].

## Data Availability

The data that support the findings of this study are openly available in OECD, https://data.oecd.org/ and also in https://kosis.kr/eng/ (accessed on 27 January 2022).

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
