# Peer review of "Inclusive Aging in Korea: Eradicating Senior Poverty"

_ijerph, 2022, doi:10.3390/ijerph19042121_

Round 1
Reviewer 1 Report
Dear authors:
I like to express my congratulations for writting an excellent and understandable piece of research regarding poverty in Korea.
My recommendation would be to clarify or emphasize the differences between rural and urban poverty differences.
In my experience, this line makes a whole difference in poverty public policy design.
Regards
Reviewer 2 Report
This is a well written descriptive policy paper dealing with a country-case study of a globally important problem of financing old-age retirement.
The paper is using a descriptive approach, without any significant use of sophisticated mathematical aparatus.
The paper is very well written. It will provide readers with a nicely presented policy discussion and a wealth of relevant informations.
I recommend the paper for publication subject to minor adjustments.
In abstract, I do not consider the formulation "strengthening the social safety net is a feasible alternative ..." as a good one. I would recommend to write that strengthening the social safety net is a was to alleviate (or contribute to alleviation of) poverty ...
On the last but one line of Abstract, i would recommend reformulate the "vocational skills development accounts" phrase since I think that we are not really talking here about any accounts (in the accounting sense).
In several places in article, the terms "we" or "our" should be replaced by Korea and Korean.
On page 3, words Chapter should be replaced by section. Chapters are in books, not in articles.
Reviewer 3 Report
This manuscript contains three main parts. The first part is a statistical description of the extent of elderly poverty in Korea. The second part addresses the dynamics of Korean elderly poverty and why policy changes are needed to mitigate a problem that will only get worse if left alone. The third part of the paper proposes three policy options. The policy options follow directly from the analysis of the problem in parts 1 and 2. All three parts of the paper are thoroughly and professionally done. The manuscript is readable by scholars in a variety of disciplines and policymakers --but some minor editing will be appropriate. The topic, analysis and discussion are entirely appropriate for the special issue. The suggestions below are not major problems.
Minor points easily fixed
The abstract does not reflect the very careful and thorough analysis of poverty and its causes in Korea. Rather, the abstract describes only the policy solutions. No one reading the abstract alone would have any idea of the careful and sophisticated analysis of poverty among the elderly in Korea.
The division of the paper into “Chapters” is awkward. This is a journal article –not a book. Use sections not chapters.
At various places in the manuscript (examples follow) it appears that the terms ‘poverty’ and ‘inequality are conflated. These terms are not the same. It is possible, for example, to have a society in which everyone is poor and inequality is either absent or very low. The reverse is also possible: A society could have a great deal of inequality and almost no one in (absolute) poverty. The sub-title of the paper and the introduction make it clear that the focus of the study is on poverty not inequality. From the first paragraph of the introduction is the following statement: “The theme of this study, the problem of poverty among the elderly, …” Yet, section 3 (page 7) is titled “Drivers of Aging Inequality in Korea and on page 8, section 3.2 is titled “Key Factors of Aging Inequality. ”And on page 15 we find the following: ”Therefore, Korea needs to expand the government's financial role to address economic inequality in old age.” The authors make it clear elsewhere that they understand the difference between poverty and inequality. My suggestion is that they examine the manuscript for instances where this distinction does not seem to be the case and correct those.
Why does table 1 contain two blank entries in the column labeled 2025. Clearly the estimates are available to fill in these two blank cells.
Appendix A contains a very useful chart which would be even more useful if it contained a breakdown of the economically inactive population.
